


# Nocturnal new particle formation events in urban environment

Farhad Salimi[1,2], Md Mahmudur Rahman[2], Sam Clifford[2], Zoran Ristovski[2] and Lidia Morawska[2]

[1]Menzies Institute for Medical Research, University of Tasmania, Hobart, Tasmania
[2]International Laboratory for Air Quality and Health, Queensland University of Technology, GPO Box 2434, Brisbane QLD, 4001, Australia

*Correspondence to:* Lidia Morawska (l.morawska@qut.edu.au)

**Abstract.** Few studies have investigated nocturnal new particle formation (NPF) events, and none of them were conducted in urban environments. Nocturnal NPF can potentially be a significant source of particles in urban areas, and studying them would improve our understanding of nucleation mechanisms. To address this, our study was conducted in an urban environment to investigate the physical characteristics of NPF events, with a particular focus on nocturnal events and the differences between them and the daytime NPF events. Particle number size distribution (PNSD) was measured for two weeks at each of 25 sites across an urban environment. A new method was proposed to automatically categorise NPF events based on growth rate (GR) in order to remove the bias related to the manual procedure. Out of 219 observed events, 118 and 101 were categorised into class I and II respectively and 73 happened during the night time which included more than 30% of the events. GR and condensation sink (CS) were calculated and a slight negative relationship between GR and CS was observed, and production of condensable vapour was lower in nighttime NPF events compared to daytime. Nocturnal events on average displayed 10% higher GRs compared to day time ones. Back trajectory analysis was also conducted to estimate the locations of the sources of day time and nocturnal precursors. While the precursors related to day time events originated from different locations with no particular pattern, back-trajectory analysis showed many air masses associated with nocturnal NPF events were transported from over the ocean. Overall, nocturnal NPF events were found to be a significant source of particles in the studied environment with different physical characteristics/sources compared to day time events.

## 1 Introduction

Atmospheric aerosols are known to affect atmospheric and climatic conditions (Solomon et al., 2007;McMurry et al., 2004) and also have an adverse effect on human health, as shown by numerous epidemiological studies (Pope II and Dockery, 2006;Dockery, 2009;Dockery and Pope, 1994;Gauderman et al., 2007). Smaller particles, such as ultrafine particles (UFPs, with a diameter < 100 nm), can have greater adverse effect on human health as they can penetrate deeply into the pulmonary system (WHO, 2005;Delfino et al., 2005;Li et al., 2003). New particle formation (NPF) events as a major source of UFPs have been observed in different types of locations around the globe including coastal, forested, mountainous, rural, and urban area (Holmes, 2007;Kulmala et al., 2004;Kerminen et al., 2010). By elevating ambient particle number concentration (PNC), NPF events can potentially affect the climate and cause adverse effects on human health. Therefore, numerous studies have investigated this phenomenon and the relevant physical properties (e.g. growth rate (GR)) and their trends around the globe.

NPF events usually occur during midday periods, indicating the photochemical origin of this phenomenon (Kulmala and Kerminen, 2008), but in some locations NPF events have also been observed during nighttime (Lee et al., 2008;Svenningsson et al., 2008;Suni et al., 2008). Man et al. (2015) found that the ammonium nitrate and organics are responsible in the nocturnal particle growth in Hong Kong. Nocturnal NPF events under low condensation sinks have been observed in the upper troposphere and from ground-based measurements (Lee et al., 2008); in contrast to day time events no





distinctive growth pattern was observed for these events (GR approximately 0 nm h$^{-1}$). Eucalypt forest was found to be an
active source of nocturnal NPF events as this phenomenon was observed in this environment in 32% of the analysed nights
in a study conducted in New South Wales, Australia (Suni et al., 2008). Chamber experiments were also conducted under
dark by varying conditions to reproduce the nocturnal events observed in the atmosphere, and it was found that in the
presence of ozone, several monoterpenes such as delta-3-carene, α-pinene, and limonene were able to produce NPF events
(Ortega et al., 2012).
Nocturnal NPF events have been studied much less than daytime events as they were usually considered as exceptions
because of the dominant theory that NPF events take place in the presence of solar radiation. However, as mentioned above,
nocturnal NPF events were found to be significant sources of particles in some environments and needs to be further studied
as understanding this phenomenon will enhance an overall knowledges of atmospheric nucleation mechanisms.  In addition,
no studies have ever reported the nocturnal events in urban environment, and there is no information available about the
characteristics of this phenomenon in more polluted area.
This work was conducted in Australia where NPF events have been previously found to be a significant contributor to the
total UFPs (Salimi et al., 2014b;Cheung et al., 2011a). This study reports for the first time on the occurrence of nocturnal
NPF events in urban environments and it aimed to determine their physical characteristics and compare them with day time
NPF events.
**2 Materials and Methods**
**2.1 Background**
From October 2010 to August 2012, air quality measurements were performed for two consecutive weeks at each of 25
randomly selected government primary schools within the Brisbane Metropolitan Area. This study was conducted within the
scope of the Ultrafine Particles from Traffic Emissions and Children's Health (UPTECH) project, which sought to determine
the relationships between exposure to traffic-related UFPs and children's health. Further details regarding the UPTECH
project and its study design can be found in  (UPTECH;Salimi et al., 2012). While this study has been performed within the
framework of UPTECH project, the results are not limited to school environments and have urban implications.
**2.2 Instrumentation, quality assurance, and data processing**
TSI Scanning Mobility Particle Sizer (SMPS) was employed to measure the PNSD within the size range 9-414 nm with five
minutes interval. TSI 3071 Differential Mobility Analyser (DMA) and a TSI 3782 water-based Condensation Particle
Counter (CPC) formed the SMPS system. Sheath flow of 6.4 lpm was supplied by employing a diaphragm pump connecting
to a critical orifice. Sheath air was dried and filtered using a silica gel dryer and a High Efficiency Particulate Air (HEPA)
filter respectively.
The SMPS system was calibrated for size accuracy using monodisperse polystyrene latex (PSL) particles, with a nominal
diameter of 100 nm, five times during the entire measurement campaign. The instruments passed all the tests with a
maximum error of 3.5% from the nominal diameter, as recommended in (Wiedensohler et al., 2012). The following quality
assurance actions were performed at each regular site visit. Sheath and aerosol flow rate of the SMPS system was measured
using a bubble flow meter. The system was zero checked by connecting the HEPA filter to the inlet of the system. Particle
loss due to diffusion was corrected using the formula derived for the laminar flow regime (Hinds, 1999). Particle loss inside





the bipolar charger and DMA was corrected using an equivalent tube length as suggested in (Karlsson and Martinsson,
2003;Covert et al., 1997).

**2.3 New particle formation identification and classification**

Surface plots of all the measured PNSD data were scanned visually for NPF events as recommended by Dal Maso et al (Dal
Maso et al., 2005). NPF events have been categorised into two main groups (Classes I and II) based on their growth rates. As
discussed by Dal Maso et al (Dal Maso et al., 2005), class I events are the ones of which the growth can be determined with
high confidence, whereas, the growth of particles in the class II events are uncertain. The criteria described in the literature
for identification of these types of events from one another are purely visual and consequently subjective. To address this
issue, a simple statistical method was developed in this study. After identifying NPF events and the period for which it was
observed, a linear regression model was fit to calculate the growth rate from the time series of count median diameter
(CMD) (Creamean et al., 2011). The linear regression model for each NPF event was thus

$$\log CMD_i = \beta_0 + \beta_1 t_i + \varepsilon_i$$

Where $\beta_0$ is the constant, $\beta_1$ is the growth rate of the CMD, $t_i$ is the slope, and $\varepsilon_i$ the residuals which are independent,
identically distributed white noise.
The 95% confidence intervals of the growth rates, $\beta_1$, were derived. For each NPF event, the growth rate and its 95%
confidence interval (CI) indicates whether an increase in CMD was observed (Class I – events with a strictly positive CI) or
not (Class II – events with a CI containing 0 and thus a growth rate indistinguishable from 0).
All statistical analysis was conducted in R (R Development Core Team, 2010).

**2.4 Condensation sink and vapour production rate**

Condensation sink (CS) is a measure of the surface area available on particles and determines the rate of condensation of
gaseous molecules on particles. CS can be calculated from particle number size distribution data, and has been used in the
literature to estimate the concentrations and source rates of condensable vapours during the NPF events (Kulmala et al.,
2005b). CS was calculated using the methods available in the literature (Pirjola et al., 1999;Lehtinen et al., 2003;Willeke,
1976;Bae et al., 2010;Salimi et al., 2014a).,
The production rate of condensable vapour was calculated using the methods in (Svenningsson et al., 2008;Kulmala and
Kerminen, 2008;Kulmala et al., 2005a).

**2.5 Kernel density estimation and generalised additive modelling**

Kernel density estimation, which is a non-parametric method to estimate the probability density function of a variable, was
employed to estimate the smoothed density of NPF events (Silverman, 1986). The relationship between the variables were
analysed using Generalised Additive Modelling (GAM) (Wood, 2003). GAM is a linear model in which the response
variable depends linearly on unknown smooth function of some explanatory variables.



**2.6 Back trajectory analysis**
In order to investigate the possible sources of the NPF events, back trajectory analysis were conducted for all the Class I
events. 24-hour air mass back trajectory was calculated using the HYSPLIT model to observe the passage of air before the
start of nucleation (Draxler and Rolph, 2003).
**3 Results and discussion**
**3.1 New particle formation events**
New particle formation events were identified by visually scanning all the surface plots and were categorised into two groups
(Classes I and II) as described in Materials and Methods. Figure 1 illustrates the calculated GRs and their 95% confidence
intervals, events that their confidence interval contains only positive values were categorised as class I, while the rest were
classified as class II events. Figure 2 shows the particles evolutions during typical Class I, where a banana shaped growth of
particles is visible, and Class II events, where burst of particles in nucleation size occurs without clear further growth.
219 events were observed in 285 days of measurements, of which 118 and 101 were categorised into Classes I and II,
respectively. The frequency of NPF events were significantly higher than previous observations in the same environment
(Cheung et al., 2010;Mejía et al., 2008) and was aligned with the results of the cluster analysis in Salimi et al. (2014b). In
our study, the apportionments of the daytime and nighttime NPF events were 67% and 33%, respectively. In this study,
overall 54.3% NPF events were class I, consisting of 34.2% daytime events and 20.1% nighttime events. GRs were ranged
between 0.015 – 13.6 (nm h$^{-1}$) during daytime and 0.25 – 11.5 (nm h$^{-1}$) during nighttime.
In our previous investigations in subtropical urban and coastal environments in the Southern Hemisphere we observed
daytime NPF events (Cheung et al., 2011b;Mejía and Morawska, 2009;Salimi et al., 2012). Daytime NPF event GRs in
Brisbane urban area, Australia were found to be in the range of 1.79 to 7.78 nm h$^{-1}$, which is comparable to this study
(Cheung et al., 2011b). Similar to our study, daytime NPF events were also observed very frequently (40% of all
observations) in urban locations in Beijing, China, during periods of low relative humidity and peak solar radiation, with the
GR of 0.1 to 11.2 nm h$^{-1}$ (Wu et al., 2007). The GRs found in our study were similar to those observed in a number of forest
sites. In particular a 10 day campaign in a Japanese forest showed midday NPFs with the particle GR between 5 and 15.7 nm
h$^{-1}$ (Han et al., 2013). In a long-term (1996-2004) measurement campaign at four Boreal forest, Finland, Dal Maso et al.
(2007) recorded GRs in the range of 0.5 –15.1 nm h$^{-1}$, which is similar to our measurement. Based on the above, it can be
concluded that daytime NPF events were observed frequently in urban areas and the reported GRs were comparable to our
study. However, nighttime NPFs were observed mostly at forest sites (Lee et al., 2008;Svenningsson et al., 2008;Suni et al.,
2008). At a forest site in Abisko, Sweden, GRs which followed nighttime NPF events were 10-40 nm h$^{-1}$ which is on average
four times higher than in our urban site study (Svenningsson et al., 2008). A rare observation of a nighttime NPF event at an
urban site in Hong Kong was recently reported by Man et al. (2015). The event was associated with particle growth higher
than in our study, ranging from, 7.1 to 39 nm h$^{-1}$, and categorised as second stage particle growth, increasing in size from
nucleation mode particles to cloud condensation nuclei (CCN) particles (61- 97 nm).




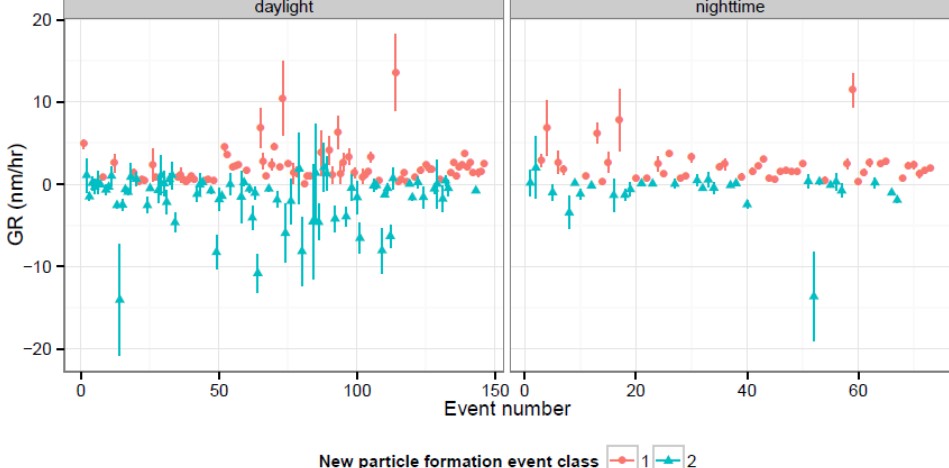


**Figure 1: Calculated GR of NPF events with their 95% CI.**

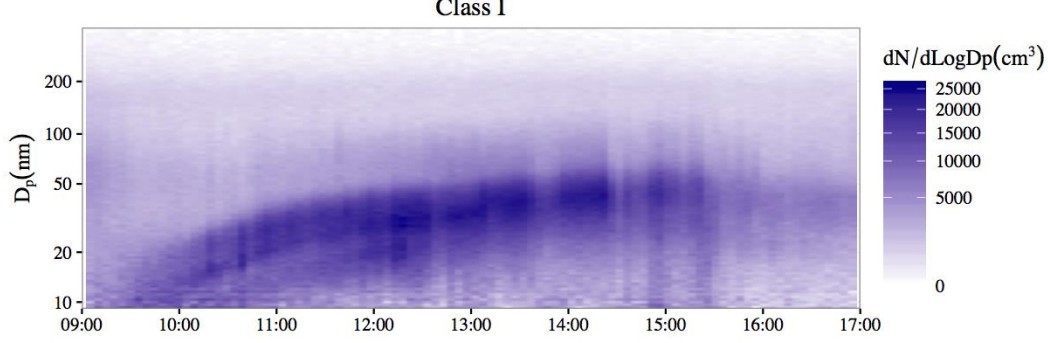

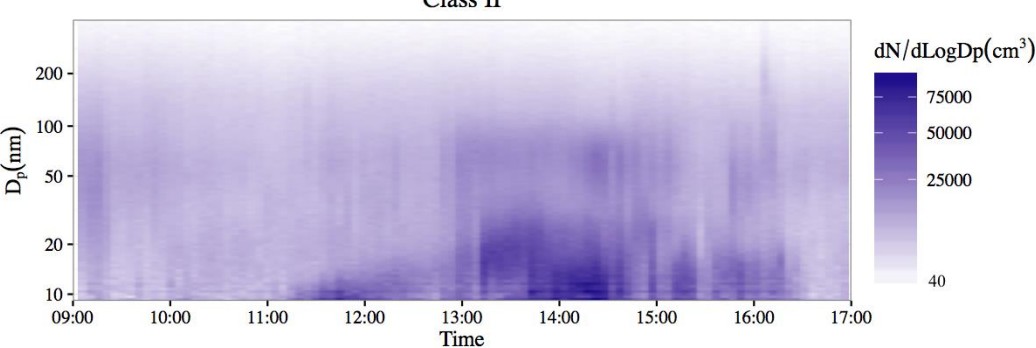


**Figure 2: Examples of the observed Class I and Class II NPF events**







## 3.2 Diurnal and temporal variation of newly formed particle growth rate

Temporal trends of GRs related to class I events were analysed using GAM as described in section 2. Diurnal model
revealed that GR had the highest value when the event started during the day light (peaking around 10 am) while night time
events were less frequent and had relatively less GR. GR had the highest and lowest values in November and May,
respectively (Figure 3). The temporal and diurnal trend analysis showed the positive correlation between the GR and the
solar radiation. The highest GR occurred during the periods with the highest solar radiation.

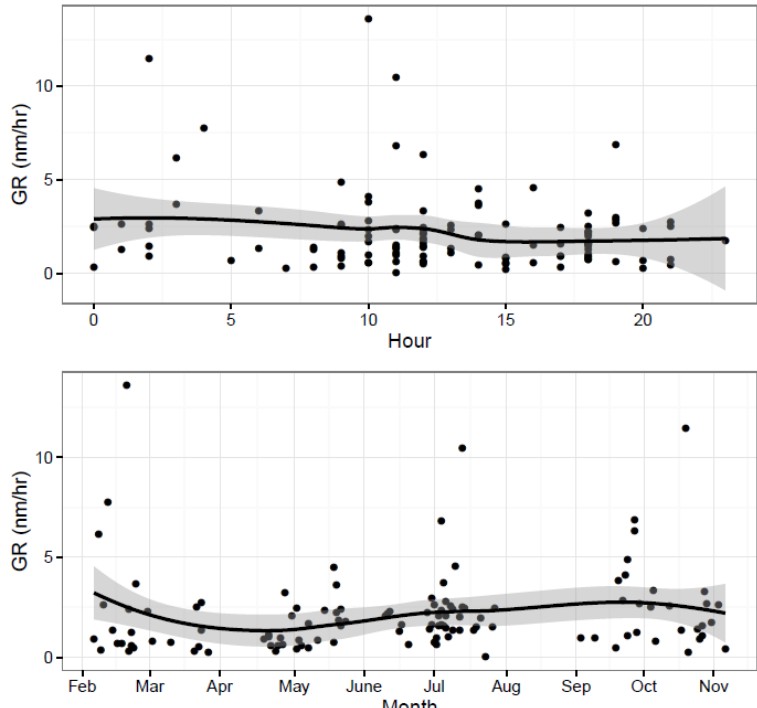


**Figure 3: Diurnal and annual trend in Growth Rate.**

## 3.3 Condensation sink and condensable vapour

The aerosol condensation sink (CS) is an important parameter that determines how fast molecules will condense onto pre-
existing aerosols (Dal Maso et al., 2002;Pirjola et al., 1999;Kulmala et al., 2005b). In this study, the calculated CS values
were averaged from the values from the period half an hour before the start of the NPF events; their relationship with the GR
was analysed using a GAM. GR is expected to be negatively correlated with CS as a higher surface area of particles leads to
higher condensation of vapour on pre-existing particles and consequently less GR (Hamed et al., 2007;Kulmala et al.,
2005b). In this study, a weak negative correlation between CS and GR was observed during both day time and nighttime
events and the uncertainties in GRs were observed in higher CS (Figure 4). However, a positive relationship between the GR
and CS, which is not clearly observed in this study, denotes the event-quenching ability of the high CS. Svenningsson et al.
(2008) concluded that high CS only allows events with high formation rate and GR to be observed as the newly formed
particles in weaker events would be scavenged by pre-existing particles. The explanation is only applicable to a high GR in
days with high CS as it cannot justify the low GR in the days with low CS (Svenningsson et al., 2008).



To investigate this further, the relationship between the calculated vapour production rate and the GR during both day and
nighttime events was modelled using GAM (Figure 4). As expected, GR showed a positive correlation with the production
rate of vapour, as more vapour results in more condensation and consequent growth. Figure 4 shows that the condensable
vapour concentration was significantly lower during night time than daytime events, indicating limited source of
condensable vapour production during night. The relationship between the CS and production rate of vapour was analysed
using GAM and a positive and negative correlations were observed during day and nighttime, respectively (Figure 4). The
positive correlation between CS and the condensable vapour, as well as the weak negative correlation between GR and CS
indicate the increase in particle emissions and vapour (e.g. vehicle and industrial emissions) during daytime. An increase in
the vapour production rate is the main cause of the increase in the GR. In case of nighttime events, it was expected that the
relationship between CS and condensable vapour would be negative due to low e emission of condensable vapour and strong
negative relationship between CS and GR.

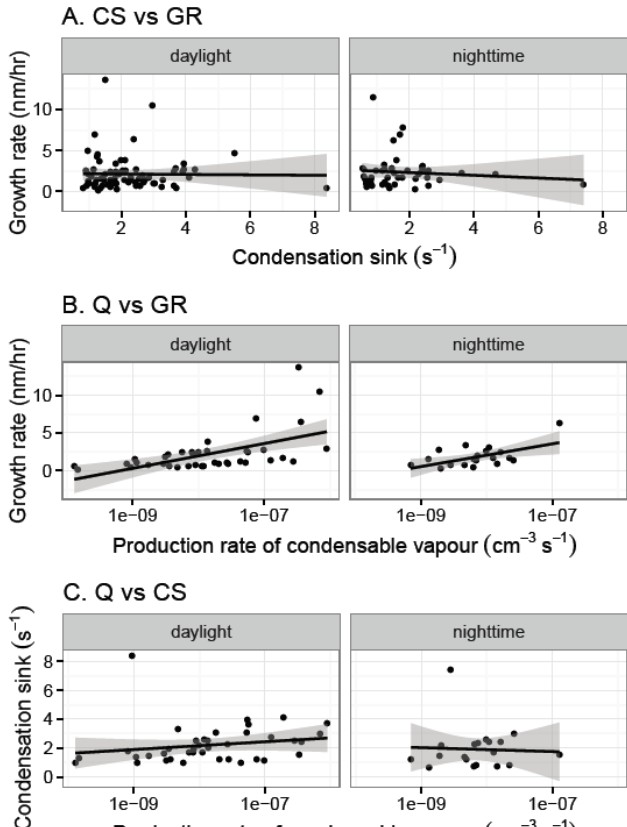


**Figure 4: Relationships between growth rate (GR), condensation sink (CS) and production rate of condensable vapour (Q) for**
**Class I nucleation evetns during daytime and nighttime. The line represents smooth trend and the shaded region represents 95%**
**confidence interval.**
**3.4 Temporal and diurnal variation of the events**
The relative frequency of the times at which NPF events occur indicate that while the bulk of the NPF events occur during
the midday period (10am-1pm), there are a number of Class I NPF events which occurred between 6pm and 7pm (Figure 5).





To investigate the unusual nocturnal events, data were divided into night time and day light based on the start of the events
using the accurate local sunrise and sunset. Out of 219 events, 73 events happened during the night time. Typical "banana
shape" in the PNSD surface plot as well as the sudden burst of newly formed particles was observed during the night time
events which are in contrast with the literature where only Class II events were observed (Lee et al., 2008) (Figure 6).
Nocturnal events occurred mostly in March and the least in December (Figure 7). On average, GRs of nocturnal events were
higher than those of day time events (Figure 8).

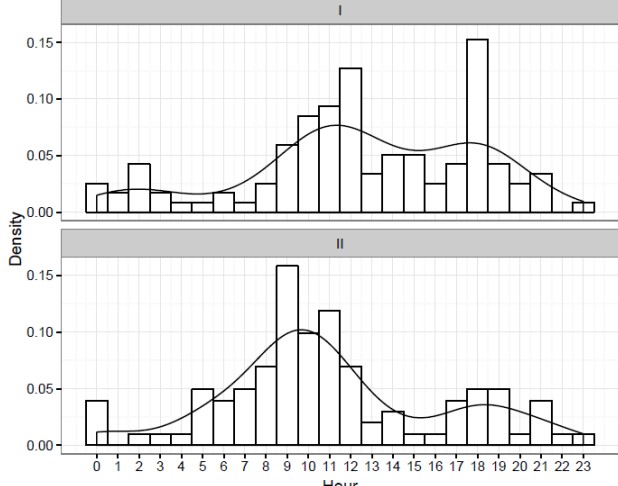


**Figure 5: Density of Class I (upper) and II (lower) NPF events with their Kernel density estimation.**

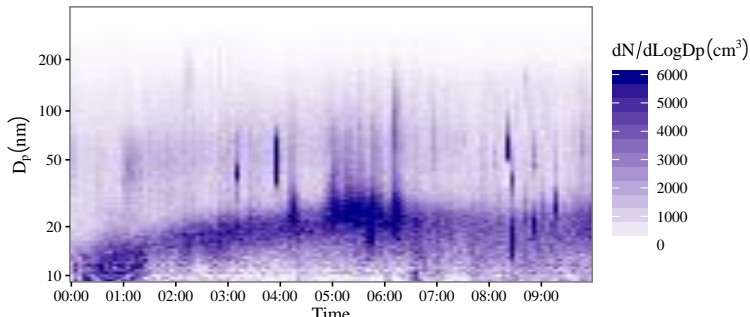


**Figure 6: An example of a banana shaped night time event.**





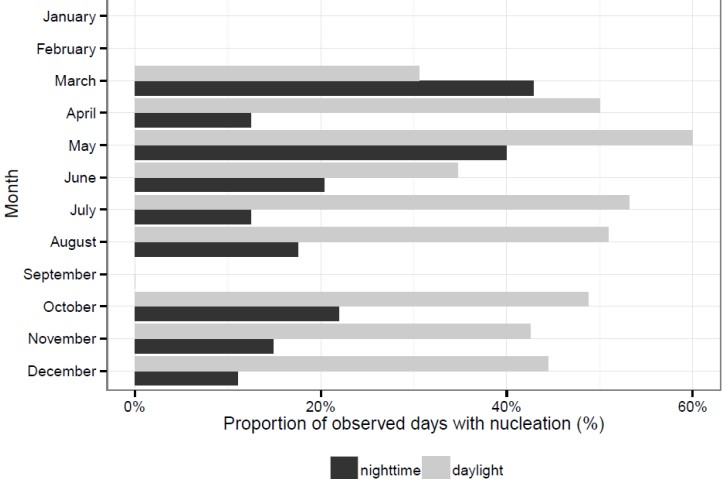


**Figure 7: Temporal trend of the night time events. Nucleation events were not observed in January, February, and September.**

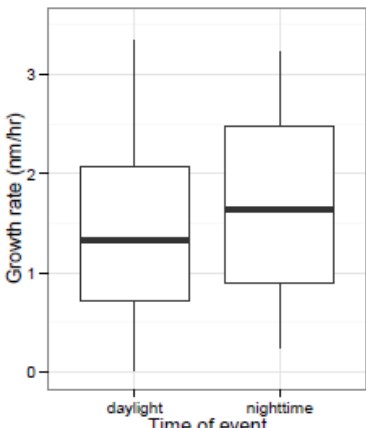


195                **Figure 8: Growth rate in day light and night time nucleation events.**

**3.5 Source of the events**
Air mass back trajectory analyses were conducted to investigate the possible sources of both day time and night time type 1
NPF events, for the 24 hours preceding the start of each event (Figure 9). The sources' locations related to daytime events
do not form a specific cluster and air masses coming from different locations seem to carry the required precursors for the
daytime events. Air mass origin is found to be an influencing factor to aerosol mass concentration, chemical composition,
and daytime NPF events in Vienna, Austria, which agrees with our findings (Wonaschutz et al., 2015). In a recent
investigation in Korea, Kim et al. (2016) found a link between daytime NPF events and continental air mass. The
relationship between nighttime NPFs and origin of air masses were not studied in those researches due to limited occurrences
of NPF events at night.
Figure 9 shows that the nighttime events were linked to air masses from the East, North-East and South-East (over the
ocean), pointing out the location of the sources of the precursors. Biogenic dimethylsulfide (DMS) compounds were





observed over the sea surface across the globe, with a higher quantities in the coral reef regions (Deschaseaux et al.,
2015;Kettle et al., 1999). With the presence of DMS, sulphur containing aerosols were observed at night in the coastal
regions in California, USA (Gaston et al., 2015). Biogenic DMS were found as a precursor of NPF in a coastal region in
Antarctica (Yu and Luo, 2010). In a recent study, Swan et al. (2016) found that emissions from coral and reef seawater are
potential sources of secondary aerosol in the Great Barrier Reef, Queensland, Australia. It is therefore possible that the
nighttime NPF identified in our study take their origin from the air mass containing biogenic oceanic precursors. To confirm
this, it is recommended that future studies would focus on comprehensive chemical characterisation of the air masses
impacting on the urban study areas.

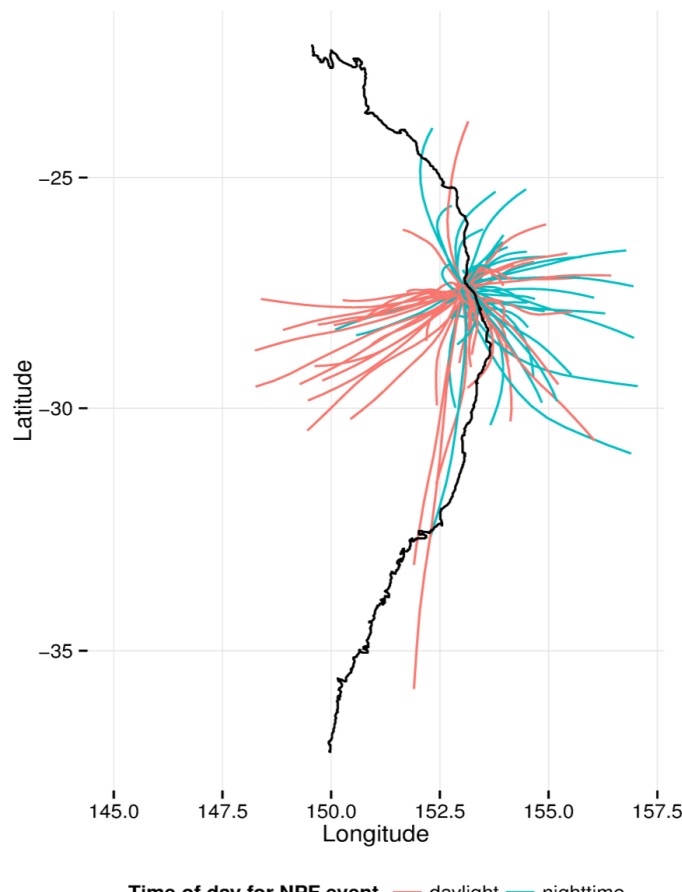


**Figure 9: 24-hr HYSPLIT back trajectory analysis for day time and night time Class I events.**
**4. Conclusion**
PNSD was measured at 25 sites within an urban environment and 219 NPF events were observed in 285 days of
measurement. A new method for classification of the events was proposed and applied successfully to the data, 118 and 169
of the events were categorised into class I and II respectively. Nocturnal NPF events were found to account for a surprisingly
high proportion (30%) of the total events. Unlike the nocturnal events observed in the literature (Lee et al., 2008), both Class



I "banana shape" and the sudden burst of newly formed particles with no growth (Class II) were observed in the PNSD
surface plot of the nocturnal events. These events occurred most commonly in March and were found to have higher GR
compared to daytime ones. CS was calculated and averaged in the period of half an hour before the start of the events, and
displayed a weak negative correlation with the GR during both day and nighttime events. However, production of
condensable vapour was lower during the nighttime events compared to day time events, as expected, and it shows negative
correlation to the nighttime CS, opposite to the daytime events, indicating different sources of precursors during daytime and
nighttime NPF events. In addition, back trajectory analysis revealed that precursors to NPF are being blown in to the
Brisbane Metropolitan Area on the East, North-East and South-Easterly, while the sources of precursors related to day time
events did not appear to display any spatial pattern. This indicates that nocturnal NPF events may have different precursors
than day time nucleation. Overall, this study found nocturnal NPF events were a significant source of ultrafine particles in an
urban environment, however, more studies need to be undertaken in order to determine the chemical characterisation of the
night time events and the chemical composition of their precursors.
**Acknowledgments**
This work was supported by the Australian Research Council (ARC), Department of Transport and Main Roads (DTMR)
and Department of Education, Training and Employment (DETE) through Linkage Grant LP0990134. Sam Clifford wishes
to acknowledge the financial support of the Institute for Future Environments (QUT) and NHRMC Centre of Research
Excellence for Air quality and health Research and evaluation.

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
