# Peer review of "Nocturnal new particle formation events in urban environment"

_Atmospheric Chemistry and Physics, 2016_

## Referee Comment (RC1) · Anonymous Referee #2 · 11 Aug 2016

This manuscripts investigates nighttime formation of new particles by nucleation in an urban environment. The analysis is based on data obtained from short measurement campaigns at 25 different sites in Brisbane, Australia. The paper contains interesting new information on nocturnal new particle formation and should therefore be published. Before acceptance, there are a number of concerns, mostly minor, that the authors should address.

Section 2.1. While detailed site information is not essential for the purpose of this paper, some information on the sites should be provided, such as which kind of different urban sites (traffic sites, urban background sites, etc.) the data set covered and how many of each site type there were.

The statement made at the end of section 2.3 (line 91) does not make any sense.

[Figure]

Please delete, or provide additional information.

Section 3.1, lines 122-137. Comparing to the whole range of observed GR values by two different studies saying that these values are similar or different not make much sense. Such a comparison should be based on narrower range of values like mean+/-STD or median+/-STD. Concerning urban areas in China, there are several recent publications from various sites to which new particle formation and growth rates could be compared as well.

Section 3.1. I do not undertand what the given CCN size range of 61-97 nm is supposed to mean here. The minimum size at which particles may as CCN depend on both supersaturation (S) and particles chemical composition. At high but realistic values of S even particles as small as 50 nm in diameter may act as CCN, whereas in most environments the minimum activation diameter is around 100 nm. There are several papers that discuss this topic.

Section 3.2, lines 145-147. The discussion is confusing. What is meant by "Diurnal model"? It seems that GAM results have somehow been applied here but the extremely short description of the model in section 2.5 makes it impossible for the reader to understand what has been really done.

Section 3.3, line 160: the event-quencing ability of high CS is an interesting observation that has been seen in many other, but not in all, earlier studies. The recent paper by Salma et al. (2016, Atmos. Chem. Phys. 16, p. 8715) investigates this same issue in detail based on an urban-rural pair of measurement data sets. The authors could discuss the role of CS a bit more and cite a couple of relevant papers.

There are some inconsistences in the values GR discussed in the text and GR shown by figures 3 and 8. While there is one high value of GR at around 10 am in Figure 3, the GR in general tends to be a bit higher at night compared with daytime. This is supported by figure 8. In the text, it is stated that GR had the highest value in November, which is not supported by figure 3.

There seems to be a scale error in the values of CS in Figure 4. Should be the real values be three orders of magnitude lower?

Language/other technical issues:

line 30: . . . urban areas

line 94: . . . onto particles

lines 98-99: incorrect way of citing papers. Should be ". . .using the method described in Svenningson et al. (2008), Kulmala. . .."

lines 120-121: the ranges should be expressed either as "in the range of M-N" or "ranged between M and N". Please correct.

line 122, . . .Hemisphere, we observed. . .

line 147: . . .less GR?

Times of days should be expressed consistently in the paper. Now the time is given as an hour of day in many figures (which is OK), while expressions like 6pm (line 181) is used in the text.

Having no space between individual papers in the reference list makes the list a bit difficult to read.

Finally, many of the figure captions contain too little information to understand the figure without search for more information from the main text. For example, the meaning of dots, lines and shadowed areas should be explained in the caption of Figure 3.
* * *

---

## Referee Comment (RC2) · Anonymous Referee #1 · 30 Oct 2016

The authors present an analysis of a dataset spanning more than one year of particle size distribution measurements that are analysed for new particle formation statistics and characteristics. Nighttime particle formation is an interesting phenomenon, and therefore the paper fits ACP in terms of its subject matter.

Overall, the paper is well structured, and the measurement description is good. There is a new method for determining the growth rate of particles, as well as classifying the events, which I consider very interesting. However, some missing description of the data analysis, as well as probable errors (detailed later) in the computations are problematic. I think that if these problems are corrected, the paper may well be published in ACP, but the mistakes and lack of description are so significant that they should be corrected.

[Figure]

The lack of description concerns the following methods:

* line 85: The method to determine the event class is very interesting and seems promising. However, some more information is needed. How is the start time of using the regression determined? What is the meaning of the constant beta_0?

* line 100, kernel density estimation: what is meant by smoothed density of NPF events? Also, I think listing the variables (both predicted and the ones used as explanatory variables) that are handled with GAM, would be beneficial to the reader.

* The whole GAM methodology remains very unclear in the paper. Especially the validity of the model for the first thing analysed seems questionable to me: the GAM result (which I'm guessing is the line in Fig 3, top panel) seems to give different results at times 0 and 24; this makes no sense at all as the diurnal cycle should not depend of the choice of start and end times. For the annual trend the same applies. This makes the whole methodology suspect, but it is difficult to identify the problem with so little explanation given.

* CS, GR and Q analysis: The obtained values seem very strange to me. The condensation sinks given (ranging from 0.5-8 $s^{-1}$) suggest that the condensing vapour lifetime is of the order of less than a second, suggesting very high concentrations of aerosol. Also, the vapour source rates are on the other hand very low, close to zero (fig 4). Unless the concentrations of aerosols are several orders of magnitude higher than usually, I think that the computations should be checked. The authors should also give in the text the variation and statistical values (mean, median, gsd, etc) of the number concentration and CS.

* Also, if the values are computed as in literature usually, it assumes a steady-state to get the formula Q-CS x C_vapour = 0, giving Q = CS x C_vapour. C_vapour is usually taken from the growth rate GR by using the free molecular regime formula C_vapour = A x GR, where A is a constant. Now, if we plot these in a x-y figure (as in figure 4), the axes are not independent, as the GR and CS are included in Q already. Therefore,

the fits in the figures are also including dependence of these variables. This should be discussed and taken into account in the analysis.

\* line 180 onwards: No clear time is given on how the daytime and nighttime were defined. This is needed for understanding the analysis.

Other comments:

\* The figure captions are much too short and it is difficult to follow what is in the figure.

\* figure 3: what are the lines and shaded areas?

\* figure 5: what is the definition of NPF density?

\* figure 9: please indicate some geographic markers (country, cities, sea, land, etc.) in the map.

One suggestion also: it would be interesting to know how the wind speed and direction affects new particle formation in the nighttime. Usually, a 'banana'-type NPF data suggests that NPF is occurring over a larger area, as advection carries air towards and away from the measurement point. However, in nighttime, wind may sometimes be non-existent. In this case, this could mean that NPF is occurring over a limited area. This could be an interesting piece of information in this case.

---

## Author Comment (AC1) · 1 Dec 2016

**Response to the reviewers' comments:**

**Reviewer 1:**

The authors present an analysis of a dataset spanning more than one year of particle size distribution measurements that are analysed for new particle formation statistics and characteristics. Nighttime particle formation is an interesting phenomenon, and therefore the paper fits ACP in terms of its subject matter.

Overall, the paper is well structured, and the measurement description is good. There is a new method for determining the growth rate of particles, as well as classifying the events, which I consider very interesting. However, some missing description of the data analysis, as well as probable errors (detailed later) in the computations are problematic. I think that if these problems are corrected, the paper may well be published in ACP, but the mistakes and lack of description are so significant that they should be corrected.

The lack of description concerns the following methods:

1.  line 85: The method to determine the event class is very interesting and seems promising. However, some more information is needed. How is the start time of using the regression determined? What is the meaning of the constant beta_0?

    Response: Authors would like to thank the reviewer for acknowledging the importance of the current work. We have added the following lines to clarify the initial time selection in the regression analysis:

    Page 4, line 88–89

    'The start and end time of the NPF events were identified visually from the surface plots of PNSD data and then incorporated in the regression analysis to calculate the growth rate.'

    The $\beta_0$ (beta_0) is the intercept of regression analysis. The term has been described now in the section 2.3 as follows:

    Page 4, line 91

    'where, $\beta_0$ is the intercept,....'

2. line 100, kernel density estimation: what is meant by smoothed density of NPF events?

Response: The smoothed density of NPF events was calculated using Kernel density estimation method and the results were presented in Figure 5. Kernel density estimation approach has been described clearly now in section 2.5 and following sentences have been added as follows:

Page 4–5, line 103–106

*'Kernel density function, also termed as Kernel smoothing, is a non-parametric method to estimate the probability density function of a variable (Silverman, 1986). Kernel smoothing is a very effective approach of visualising data structures without incorporating parametric model (Wand and Jones, 1994). In this study, Kernel smoothing estimated the overall diurnal trend of the NPF events over the study period in conjunction with the histogram plot. We applied 'ggplot2' package in R programming language to plot smooth Kernel density trend of NPF events ([www.ggplot2.org](www.ggplot2.org)).'*

3. Also, I think listing the variables (both predicted and the ones used as explanatory variables) that are handled with GAM, would be beneficial to the reader.
Response: The following sentences have been added:

Page 5, line 111 –112
'One explanatory variable was used in each model. Hour, month, condensation sink, and production rate of the condensable vapor were used as the explanatory variables in each model.'

4. The whole GAM methodology remains very unclear in the paper. Especially the validity of the model for the first thing analysed seems questionable to me: the GAM result (which I'm guessing is the line in Fig 3, top panel) seems to give different results at times 0 and 24; this makes no sense at all as the diurnal cycle should not depend of the choice of start and end times. For the annual trend the same applies. This makes the whole methodology suspect, but it is difficult to identify the problem with so little explanation given.

Response: GAM has been explained in the section 2.5 and following sentences have been added in the revised manuscript as below:

Page 5, line 107 – 110

*'The mathematical framework in GAM is similar to the Generalized Linear Model (GLM); however, it replaces the linear function with non-parametric smoothers (e.g., penalised splines), which allow for flexible estimation of non-linear function. GAMs were found more appropriate than GLMs in estimating non-linear effect of response variables in air quality study (Clifford et al., 2011).'*

Regarding Figure 3, we would like to explain that we used 0-23h time instead of 01-24 h, which

means the diurnal GR values were plotted in times between 0 h and 23 h. Therefore, there is no overlap between times. Similarly, there is no overlap in annual trends. Figure 3 and its caption have been modified to:

[Figure]

Figure 3: Diurnal (0 h to 23 h) and annual (March to December) trend in Growth Rate. NPF events were not observed in January and February. The solid line represents smoothed trend modelled with GAM and the shaded region represents 95% confidence interval.

5.  CS, GR and Q analysis: The obtained values seem very strange to me. The condensation sinks given (ranging from 0.5-8 s^(-1)) suggest that the condensing vapour lifetime is of the order of less than a second, suggesting very high concentrations of aerosol. Also, the vapour source rates are on the other hand very low, close to zero (fig 4). Unless the concentrations of aerosols are several orders of magnitude higher than usually, I think that the computations should be checked. The authors should also give in the text the variation and statistical values (mean, median, gsd, etc) of the number concentration and CS.

Response: We have checked the analysis of GR and Q and indeed found an error in the programming code in R. CS and Q have been recalculated and Figure 4 have been updated as follows:

[Figure]

[Figure]

6. Also, if the values are computed as in literature usually, it assumes a steady-state to get the formula Q-CS x C_vapour = 0, giving Q = CS x C_vapour. C_vapour is usually taken from the growth rate GR by using the free molecular regime formula C_vapour = A x GR, where A is a constant. Now, if we plot these in a x-y figure (as in figure 4), the axes are not independent, as the GR and CS are included in Q already. Therefore, the fits in the figures are also including dependence of these variables. This should be discussed and taken into account in the analysis.

Response: We agree with the reviewer that two graphs (GR vs. CS and CS vs. Q) in Figure 4 are enough as the third graph (Q vs. GR) is not independent of the other two; so we decided to keep GR vs. CS and CS vs. Q. We have removed the Q vs CS section of the graph and the releted discussion from the paper. The discussion is now limited to two graphs (i.e. GR vs. CS and CS vs. Q).

7. line 180 onwards: No clear time is given on how the daytime and nighttime were defined. This is needed for understanding the analysis.

Response: Accurate local sunrise and sunset time were used in this study and this information has been added in the sentence below:

Page 10, line 193 – 194

*'To investigate the unusual nocturnal events, data were divided into nighttime and daylight based on the start of the events using the accurate local sunrise and sunset time.'*

8. Other comments: The figure captions are much too short and it is difficult to follow what is in the figure.

Response: The captions have been revised to include more information:

*'Figure 1: Calculated growth rate (GR) of day light and nighttime NPF events with their 95% confidence interval.'*

*'Figure 2: Examples of the observed Class I and Class II NPF events. $D_p$ is particle diameter and the colour of the image represent measured concentrations at each time.'*

*'Figure 3: Diurnal (0 h to 23 h) and annual (March to December) trend in growth rate (GR). GR was not observed in January and February. The line represents smoothed trend modelled with GAM and the shaded region represents 95% confidence interval.'*

*'Figure 4: Relationships between growth rate (GR), condensation sink (CS) and production rate of condensable vapour (Q) for Class I NPFs during day light and nighttime. The line represents smoothed trend modelled with GAM and the shaded region represents 95% confidence interval.'*

*'Figure 5: Diurnal trend of Class I and II NPF events with their Kernel density estimation.'*

*'Figure 6: An example of a banana shaped nighttime event. Dp is particle diameter and the colour of the image represent measured concentrations at each time.'*

9. figure 3: what are the lines and shaded areas?

Response: Please see the response to the comment 8.

10. figure 5: what is the definition of NPF density?

Response: Figure 5 shows the diurnal trend of NPF events with Kernel density estimation. The caption of Figure 5 has been modified as below:

*'Figure 5. Diurnal trend of Class I and II NPF events with their Kernel density estimation.'*

11. figure 9: please indicate some geographic markers (country, cities, sea, land, etc.) in the map.

Response: Figure 9 has been replotted as follows:

[Figure]

Time of day for NPF event —— daylight —— nighttime

12. One suggestion also: it would be interesting to know how the wind speed and direction affects new particle formation in the nighttime. Usually, a 'banana'-type NPF data suggests that NPF is occurring over a larger area, as advection carries air towards and away from the measurement point. However, in nighttime, wind may sometimes be non-existent. In this case, this could mean that NPF is occurring over a limited area. This could be an interesting piece of information in this case.

Response: Thank for your suggestion, however, the investigation of the effects of wind speed and its direction to the NPFs is not within the scope of current paper and will be investigated in future work.

**Reviewer 2:**

This manuscript investigates nighttime formation of new particles by nucleation in an urban environment. The analysis is based on data obtained from short measurement campaigns at 25 different sites in Brisbane, Australia. The paper contains interesting new information on nocturnal new particle formation

and should therefore be published. Before acceptance, there are a number of concerns, mostly minor, that the authors should address.

1.  Section 2.1. While detailed site information is not essential for the purpose of this paper, some information on the sites should be provided, such as which kind of different urban sites (traffic sites, urban background sites, etc.) the data set covered and how many of each site type there were.

    Response: A brief description on the study sites has been added in the main text, as below:

    Page 3, line 58-63
    *"All 25 sites were located between 1.5 and 30 km from Brisbane city. Some sites were affected more by high traffic density than others. The average hourly traffic counts at the nearby roads to the sites were ranged between 44 to 1217 (Laiman et al., 2014). Further details regarding the UPTECH project and its study design can be found in our previous publication (Salimi et al., 2013)"*

2.  The statement made at the end of section 2.3 (line 91) does not make any sense. Please delete, or provide additional information.

    That sentence has been reworded to:

    Page 4, line 93–94
    *'For each NPF event, growth rate (GR) and its related 95% confidence interval (CI) were calculated. When the CI's were positive, they were classified as Class I, and the rest of events were classified as Class II.'*

3.  Section 3.1, lines 122-137. Comparing to the whole range of observed GR values by two different studies saying that these values are similar or different not make much sense. Such a comparison should be based on narrower range of values like mean+/- STD or median+/-STD. Concerning urban areas in China, there are several recent publications from various sites to which new particle formation and growth rates could be compared as well.

    Response: In the revised manuscript (section 3.1), we have compared the average GRs and here also included a recent study that calculated GRs in China. The modified section reads as seen:

    Page 5–6, line 131–149
    *"Day light NPF event average GRs obtained by author of an earlier study conducted also in Brisbane (Cheung et al., 2011), were found to be of 4.6 nmh$^{-1}$ (range of 1.79 to 7.78 nm h$^{-1}$), which is one order of magnitude higher than in this study (2.4 nm h$^{-1}$). Cheung et al. (2011) calculated GRs based on a long-term measurement data at a single site; however this study used*

*data from 25 sites to calculate average GRs and therefore GRs in this study are expected to be more representative of the area of the study than the former one.*

*Studies from the other places in the world show higher GRs in some cases but also similar to this study. Similarly to our study, day light NPF events were also observed frequently (40% of all observations) in urban locations in Beijing, China, during periods of low relative humidity and peak solar radiation, with the average GRs of 1.8 and 4.4 nm h$^{-1}$ during clean and polluted NPF events, respectively (Wu et al., 2007). In a recent study in North China Plain, Shen et al. (2016) found day light NPF event average GRs of 1.2 nm h$^{-1}$ higher than in this study. A 10-day campaign in a Japanese forest showed midday NPFs with the mean particle GR of 9.2 nm h$^{-1}$, ranged between 5 and 15.7 nm h$^{-1}$, which is approximately four times higher than our study (Han et al., 2013). However, in a long-term (1996-2004) measurement campaign at four Boreal forest, Finland, Dal Maso et al. (2007) recorded the average GRs of 3.0 nm h$^{-1}$ (range of 0.5 –15.1 nm h$^{-1}$), which is similar to this in our study. However, nighttime NPFs were observed mostly at forest sites (Lee et al., 2008;Svenningsson et al., 2008;Suni et al., 2008). At a forest site in Abisko, Sweden, GRs which followed nighttime NPF events were 10-40 nm h$^{-1}$ which is on average four times higher than in our urban site study (Svenningsson et al., 2008). A rare observation of a nighttime NPF event at an urban site in Hong Kong was recently reported by Man et al. (2015). The event was associated with particle growth, and the GRs were higher than in our study, ranging from 7.1 to 39 nm h$^{-1}$.”*

4. Section 3.1. I do not understand what the given CCN size range of 61-97 nm is supposed to mean here. The minimum size at which particles may as CCN depend on both supersaturation (S) and particles chemical composition. At high but realistic values of S even particles as small as 50 nm in diameter may act as CCN, whereas in most environments the minimum activation diameter is around 100 nm. There are several papers that discuss this topic.

Response:

That sentence has been reworded to:

Page 6, line 148

*'The event was associated with particle growth higher than in our study, ranging from, 7.1 to 39 nm h$^{-1}$.'*

5. Section 3.2, lines 145-147. The discussion is confusing. What is meant by "Diurnal model"? It seems that GAM results have somehow been applied here but the extremely short description of the model in section 2.5 makes it impossible for the reader to understand what has been really done.

Response:

The GAM model descrition has been revised in the section 2.5. Please see response to comment 4 of the reviewer 1.

In addition, the section 3.2 has been revised to:

Page 7, line 157–162

*'GAM model fit to the diurnal GR data revealed that GR had the highest value when the event started during the day light (peaking around 10 am) while nighttime events were less frequent and had relatively lower GR (Figure 3). GAM model fit shows that the GR had the highest and lowest values in October and May, respectively (Figure 3). The temporal and diurnal trend analysis showed positive correlation between the GR and the solar radiation. The highest GR occurred during the periods with the highest solar radiation.'*

6. Section 3.3, line 160: the event-quencing ability of high CS is an interesting observation that has been seen in many other, but not in all, earlier studies. The recent paper by Salma et al. (2016, Atmos. Chem. Phys. 16, p. 8715) investigates this same issue in detail based on an urban-rural pair of measurement data sets. The authors could discuss the role of CS a bit more and cite a couple of relevant papers.

Response: The recent paper by Salma et al. (2016) has been added in the updated manuscript. To clarify the role of CS in NPF events, the following sentences have been added in the Section 3.3:

Page 8–9, line 178–181

*"Salma et al. (2016) observed particle quenching/shrinkage events in NPFs which were linked to atmospheric conditions in Budapest, Hungary. The study observed 25% decrease in CS concentration during shrinkage phase compared to growth phase. Similar findings were observed in Po Valley, Italy (Hamed et al., 2007)."*

7. There are some inconsistences in the values GR discussed in the text and GR shown by figures 3 and 8. While there is one high value of GR at around 10 am in Figure 3, the GR in general tends to be a bit higher at night compared with daytime. This is supported by figure 8. In the text, it is stated that GR had the highest value in November, which is not supported by figure 3.

Response: The relevant description in section 3.2 has been modified as explained above in the response 5.

GAM model fit to the monthly GR data revealed their annual trend, as seen in Figure 3. The

following sentence has been added in section 3.2:

Page 7, line 159

*"GAM model fit shows that the GR had the highest and lowest values in October and May, respectively (Figure 3)".*

Also we have mentioned the findings of overall median GRs during day light and nighttime events in the section 3.4 (Figure 8), as below:

Page 10, line 197

*"On average, GRs of nocturnal events were higher than those of day light events (Figure 8)"*

8. There seems to be a scale error in the values of CS in Figure 4. Should be the real values be three orders of magnitude lower?
   Response: Authors would like to thank the reviewer for noticing this problem. Yes, there was an error in the programming code in R. We have reanalysed the whole data and calculated the CS and Q again. Figure 4 has been plotted again.

Language/other technical issues:

1. line 30: . . . urban areas
   Response: Fixed
2. line 94: . . . onto particles
   Response: Fixed
3. lines 98-99: incorrect way of citing papers. Should be ". . .using the method described in Svenningson et al. (2008), Kulmala. . .."
   Response: Fixed
4. lines 120-121: the ranges should be expressed either as "in the range of M-N" or "ranged between M and N". Please correct.
   Response: Fixed
5. line 122, . . .Hemisphere, we observed. . .
   Response: Fixed
6. line 147: . . .less GR?
   Response: Fixed
   Times of days should be expressed consistently in the paper. Now the time is given as an hour of day in many figures (which is OK), while expressions like 6pm (line 181) is used in the text.
   Response: Fixed
7. Having no space between individual papers in the reference list makes the list a bit difficult to

read.

Response: Fixed

8. Finally, many of the figure captions contain too little information to understand the figure without search for more information from the main text. For example, the meaning of dots, lines and shadowed areas should be explained in the caption of Figure 3.

Response: Please see the response to comment 8 of the reviewer 1.